# Expert consensus on pre-eclampsia risk screening tools for low- and middle-income countries: Development of a new Target Product Profile

Annie R. A. McDougall[1,2,*], Tahlia R. Guneratne[1], Kate Mills[1], Maureen Makama[1,2], Ahmet Metin Gülmezoglu[3,4], Anne Ammerdorffer[3,4], Lindsay Keir[5], Cecile Ventola[5], Jennifer Scott[3,4], Marta Fernández-Suárez[3,4], Joshua P. Vogel[1,2]

1 Women's, Children's and Adolescent's Health Program, Burnet Institute, Melbourne, Australia, 2 School of Public Health and Preventive Medicine, Monash University, Melbourne, Australia, 3 Concept Foundation, Geneva, Switzerland, 4 Concept Foundation, Bangkok, Thailand, 5 Impact Global Health, Sydney, Australia

☯ These authors contributed equally to this work.
* annie.mcdougall@burnet.edu.au

## Abstract

Risk screening tools can identify women at high-risk for developing pre-eclampsia, allowing for timely preventative care and monitoring. However, tools that are accurate, feasible and affordable for limited-resource settings are lacking. Target Product Profiles (TPPs) define the minimum and optimal criteria required for new products to achieve a specific health need. Through the Accelerating Innovation for Mothers (AIM) project, we developed the first TPP for pre-eclampsia risk screening tools, suitable for use in low- and middle-income countries (LMICs). We used a mixed-methods approach, including stakeholder interviews, an online multilingual survey, and public consultation to inform the draft TPP. Diverse stakeholders (e.g., clinicians, manufacturers, researchers) representing all WHO geographical regions were invited to participate. We used descriptive statistics to identify variables that met consensus (≥75% agree or strongly agree) from surveys and qualitative content analysis to identify emerging themes from interviews. We conducted 26 interviews and received 84 survey responses, from diverse participants in terms of geographies, gender and country income level. Consensus was reached on all but two of the 25 TPP variables (tool recommendations, sample collection). Following interview feedback, we modified the TPP to include three distinct sections (biophysical, biochemical and digital), allowing these tool types to be clearly defined. This is the first TPP on risk screening tools for predicting pre-eclampsia. This will facilitate the development, evaluation and advancement of pre-eclampsia risk screening tools for pregnant women. It can help ensure these tools address real-world needs and current gaps in pre-eclampsia related care in LMICs.

**Data availability statement:** All data is included within the manuscript and Supporting information.

**Funding:** The research was funded by The Gates Foundation (Grant INV-038938). JPV is supported by an Australian National Health and Medical Research Council Emerging Leadership Investigator Grant.(2020/GNT1194248). ARAM is supported by the Lady Potter Emerging Leader Fellowship (Burnet Institute). The funders had no role in the study design, data collection and analysis, decision to publish, or preparation of the manuscript.

**Competing interests:** The authors have declared that no competing interests exist.

## Introduction

Pre-eclampsia is a complication of pregnancy characterised by new-onset hypertension, proteinuria and/or end organ dysfunction, presenting at or after 20 weeks' gestation [1]. Globally, an estimated 4.6% of pregnant women will experience pre-eclampsia annually [2]. Hypertensive disorders of pregnancy are the second-leading cause of maternal deaths globally, the majority these are due to pre-eclampsia or eclampsia [3]. They also contribute significantly to fetal and neonatal mortality and morbidity, particularly fetal growth restriction, stillbirth and preterm birth [4]. This disease has its origins in abnormal placental development, though the main trigger and the subsequent biological cascades are not yet fully understood [5]. A number of immunologic, genetic, environmental, obstetric, medical and sociodemographic risk factors are associated with its occurrence [6].

Tools that can accurately identify women at risk of pre-eclampsia in early pregnancy are essential, as they allow timely initiation of preventive therapies (such as low-dose aspirin for high-risk women), as well as enhanced antenatal monitoring [7]. Pre-eclampsia risk screening has traditionally relied on a 'checklist' approach, where women with certain risk factors, usually from patient history, are treated as high risk [8,9]. This approach has shown relatively poor accuracy [10,11]. For example, a prospective study of the National Institute for Clinical Excellence (NICE) guideline's risk screening approach showed it identified only 39% of women who went on to develop preterm pre-eclampsia, and 34% who developed term pre-eclampsia [12]. If early pregnancy risk screening has poor accuracy, many women who will develop pre-eclampsia are not identified, nor offered the necessary preventive care.

Newer tools, such as the Fetal Medicine Foundation (FMF) Bayesian algorithm, combine clinical risk factors with biomarker tests, including uterine artery pulsatility index (UtA-PI), mean arterial pressure (MAP) and serum biomarkers [13]. The FMF tool has shown superior predictive performance [14], yet routine biomarker testing is difficult to implement in many settings [15]. This requires laboratory and medical imaging services to be routinely available wherever antenatal care is delivered. Yet for many health services internationally, inadequate infrastructure, limited supplies, and shortages of health and laboratory staff are a daily reality [16,17]. Similarly, the FMF tool was validated for use at 11 to <14 weeks' gestation, yet in low-income countries only 24% of women start antenatal care in the first trimester [18]. Many countries continue to use traditional history-based risk screening, though it has limited predictive performance [11,12].

The Accelerating Innovation for Mothers (AIM) project, established in 2020, seeks to encourage research and development (R&D) investment for pregnancy-related conditions [19]. As part of AIM, we identified 89 pre-eclampsia risk screening tools in the R&D pipeline [20]. Most of this research has focused on new maternal blood biomarkers, which will likely face similar implementation challenges to current biomarker-based testing technologies. We consider it unlikely that the current pipeline of technologies will adequately address the needs and realities of low- and middle-income countries (LMICs).

One way to improve the R&D landscape is to develop a Target Product Profile (TPP). TPPs are strategic documents that define the minimum and optimal characteristics of new health products. They guide key stakeholders, including funders and product developers, on how to best meet clinical and public health needs [21]. Within AIM, we produced the first maternal TPPs for novel medicines to prevent and treat pre-eclampsia [22–24], prevent preterm birth and manage preterm labour [25–27], and optimise maternal gut microbiome [28]. We aimed to develop the first publicly available TPP for pre-eclampsia risk screening tools, to help improve pre-eclampsia prevention and mitigate the effects of pre-eclampsia globally, especially in LMICs where the burden is greatest.

## Materials and methods

### Ethics statement

The study protocol was reviewed and approved by the Alfred Ethics Committee for Human Research (Project number 100/24). A formal statement of consent was given by all participants, both written and verbally (interview participants) or electronically (survey participants), as approved by the Alfred Ethics Committee for Human Research. Interviewees first provided written consent to participate in an interview via response to the email invitation. Their consent was confirmed verbally during the interview and recorded in the interview notes.

### Target product profile development

The study protocol was based on our previous TPP development for maternal medicines (Fig 1), [24,29]. The structure of the TPP was informed by existing TPPs from the AIM project, the World Health Organization (WHO) and other international organisations [24,29,30]. The initial TPP draft was developed based on extensive literature review and discussions with maternal health and pre-eclampsia subject matter experts. The draft outlined 25 key variables and their minimum and optimal targets, as well as the intended use case scenario, which was revised in subsequent phases (Box 1 presents the final version). We conducted a comprehensive stakeholder mapping process to ensure diverse participation from ten different stakeholder groups: obstetricians, midwives/nurses, academic researchers, antenatal care program managers, maternal diagnostics manufacturers, consumer representatives, guideline panel members, procurement experts, global diagnostics and innovation representatives, and international health agency/organisation staff. Stakeholders were identified through systematic searching and snowball sampling (S1 Appendix).

**Individual stakeholder interviews and public consultation.** From this mapping, we sampled stakeholders to invite for an interview – this sampling ensured diversity across geographies, country income level, and gender. One-on-one interviews were conducted online by one of our research team using a semi-structured interview guide (S2 Appendix). The interviews sought feedback on the draft TPP, focusing on the intended use case scenario and the minimum and optimal targets for each variable. For those agreeing to an interview, a copy of the draft TPP was provided prior. Interviews lasted 45–65 minutes and were conducted in English via Microsoft Teams. Informed consent was obtained at the beginning of each interview, including consent to record the interviews (to supplement notes captured during the interview). Interviews were conducted until thematic saturation was achieved, with ongoing analysis used to assess the emergence of new themes. Recruitment also ensured that all identified stakeholder groups were represented, even where saturation within a particular group was reached earlier.

**International stakeholder survey.** Concurrent with the interviews, an international online stakeholder survey was conducted using the survey platform Qualtrics [31]. Respondents rated their agreement or disagreement with the intended use case scenario and targets for each variable (minimum and optimal) using a 5-point Likert scale (S3 Appendix). Consensus for each variable was defined as at least 75% of respondents either agreeing or strongly agreeing. Free-text comments could be provided for each variable. The survey and invitation email were available in English, French and Spanish.

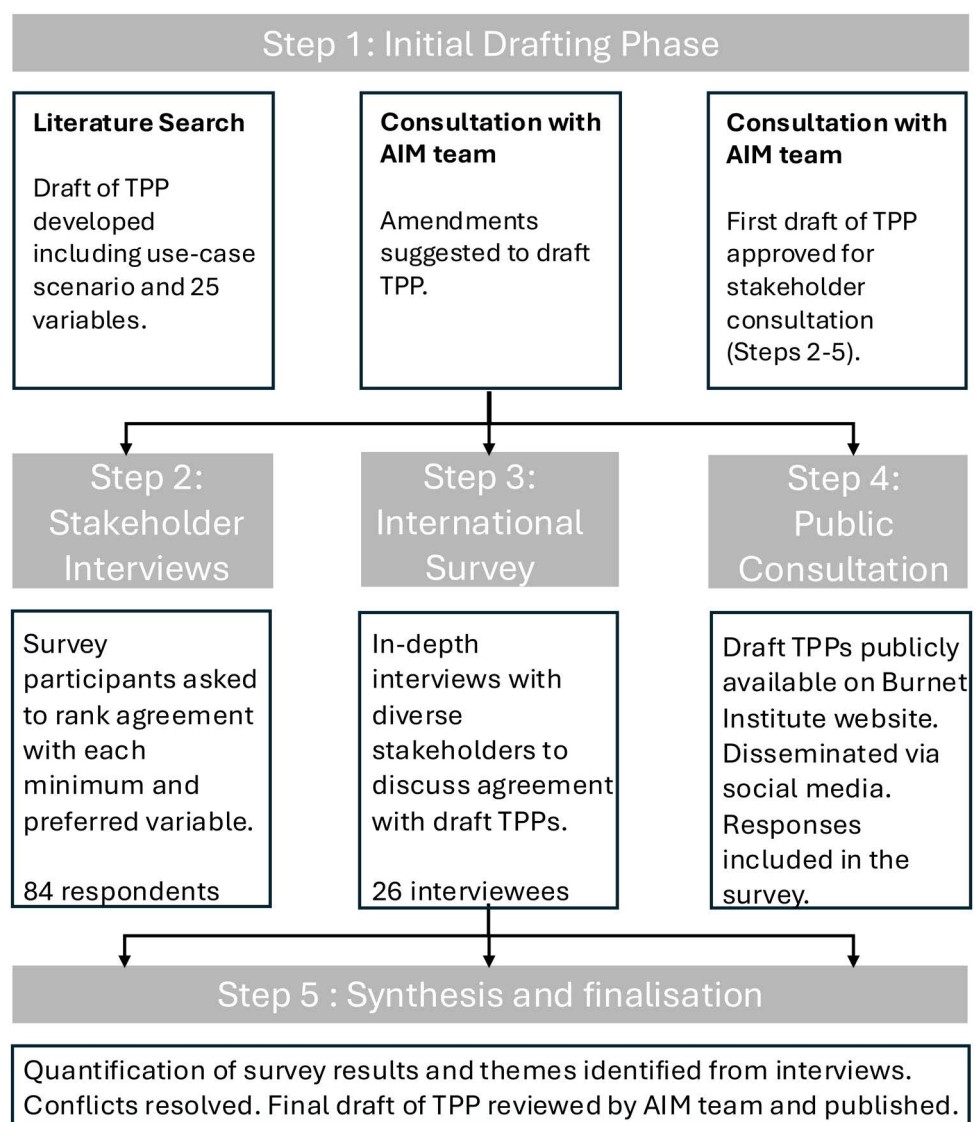

**Fig 1. TPP development process overview.**

Survey invitations were sent via email to all key stakeholders we identified across all ten stakeholder groups. The survey was also shared on social media and communication platforms of various societies and organisations (S1 Appendix). Survey completion was anonymous, and recipients were encouraged to share the invitation with their networks, so an exact response rate could not be calculated.The draft TPP was published online on the Burnet Institute and Concept Foundation websites for public comment, along with the survey link.

**Synthesis, analysis and finalisation.** Interviews were analysed using a qualitative content analysis approach [32]. This was intended to identify the significant points raised for each TPP variable, as well as potential development and implementation barriers. Analysis was performed by two researchers (TG and AM) using Microsoft Excel to collate findings. Disagreements were resolved via discussion, merging or splitting codes, and if required, discussion with a

> **Box 1. Final use case scenario in the TPP for pre-eclampsia risk screening tools.**
>
> A screening tool that can be used to accurately identify pregnant women at high (or increased) risk of pre-eclampsia. The tool is used for the prediction of pre-eclampsia, not diagnosis, and therefore the target population is pregnant women without a diagnosis of pre-eclampsia. Identification of women at increased risk of developing pre-eclampsia allows for referral, more frequent clinical monitoring as well as timely commencement of preventive interventions (such as low-dose aspirin), to prevent the onset of pre-eclampsia.
>
> The tool will be suitable for use during pregnancy as part of routine antenatal care, in low- and middle-income countries. It will be used by health personnel who deliver antenatal care services and will be user-friendly and affordable across a range of settings.
>
> The TPP is separated into three tables to outline the different properties of the tool.
>
> - General characteristics of the tool. These TPP variables will apply to all pre-eclampsia risk screening tools, regardless of the technology or method used.
>
> - Three potential components of the risk screening tool (Biochemical tests, Biophysical tests and Digital platform). These TPP variables will only apply to those pre-eclampsia risk screening tools that include these components.
>
> - The data management requirements of the tool. These TPP variables will apply to all pre-eclampsia risk screening tools which collect and use data.

third researcher (KM and MFS). A combined directed and summative content analysis approach was utilised. A directed approach was used to develop an initial coding framework based on the TPP variables, our previous experience developing TPPs and knowledge of the existing literature. A summative approach was then used to identify, count and summarise the codes into key themes, split into major and minor points of feedback. The themes were analysed to determine the levels of agreement or disagreement by interviewees within each TPP variable. The interview findings were triangulated with the survey findings to ensure rigour.

Survey responses were analysed descriptively, to determine whether consensus was reached for each variable. We defined consensus as 75% or more respondents agreeing to the variable. Survey respondents that only completed the demographic data were excluded from analysis. All written comments submitted in the survey were translated (if required), reviewed, and particular attention given to comments on variables that did not reach consensus. All feedback were collated, discussed by the research team and informed subsequent TPP revisions. When variables did not reach consensus, and/or there was disagreement among stakeholders, feedback from stakeholders in LMICs was prioritised, particularly for input related to ease of use and scale-up. Input from stakeholders with a role in tool development (researchers, product manufacturers) related to the structure of the TPP (as opposed to the content) was prioritised. The final TPP version was reviewed by the AIM project team before finalisation and publication.

## Results

### Individual stakeholder interviews

Between 12 June and 6 September 2024, we contacted 85 stakeholders, of which 26 agreed to be interviewed (18 females and 8 males), with representation from all WHO regions (Table 1). Of these, 54% were from LMICs and 46% from

Table 1. Distribution of stakeholders by WHO global region and gender.

| Stakeholder group* | Interview participants n = 26 (%) | Survey participants n = 84 (%) |
|---|---|---|
| Obstetricians | 4 (15.4) | 17 (15.5) |
| Midwives/Nurses | 2 (7.7) | 21 (18.1) |
| Academics | 3 (11.5) | 26 (22.4) |
| Global diagnostic and innovation experts | 3 (11.5) | 3 (2.6) |
| Antenatal care program managers | 3 (11.5) | 5 (4.3) |
| Maternal Diagnostic manufacturers | 3 (11.5) | 3(2.6) |
| Guideline panel members | 3 (11.5) | 5 (4.3) |
| Procurement expert | – | 1 (0.9) |
| International health organisation staff | 3 (11.5) | 15 (12.9) |
| Consumer representative | 2 (7.7) | 3 (2.6) |
| Other | – | 16 (13.8) |
| **Gender** | | |
| Woman | 18 (69.2) | 55 (65.5) |
| Man | 8 (30.8) | 27 (32.1) |
| Non-binary/gender diverse | – | 1 (1.3) |
| Prefer to self-describe | – | 1 (1.3) |
| **Geographic location** | | |
| African region | 6 (23.1) | 28 (33.3) |
| Region of the Americas | 4 (15.4) | 14 (16.7) |
| Eastern Mediterranean region | 1 (3.8) | 2 (2.4) |
| European region | 5 (19.2) | 17 (20.2) |
| South-East Asian region | 2 (7.7) | 2 (2.4) |
| Western Pacific region | 8 (30.8) | 21 (25.0) |
| **Country income level** | | |
| High income | 12 (46.1) | 39 (46.4) |
| Upper middle income | 11 (42.3) | 8 (9.5) |
| Lower middle income | 2 (7.7) | 31 (36.9) |
| Low income | 1(3.9) | 6 (7.1) |

* The survey participants could select more than one stakeholder group relevant to them, therefore the total number for this group exceeds 84.

high-income countries (HICs). Complete list of countries represented by stakeholders described in S4 Appendix. Nine key stakeholder groups were captured in these interviews; we were unable to identify a procurement expert willing to be interviewed.

**Intended use of tool.** Interview feedback was consistent across participants from both HICs and LMICs. Participants agreed with the need for pre-eclampsia risk screening tools. Major themes included adjusting the 'use case scenario' and 'intended use' to specify that the tool should not be used for women with suspected or diagnosed pre-eclampsia (S5 Appendix). Participants agreed that the purpose of the tool should be risk screening, emphasising the need to predict any pre-eclampsia, not only preterm pre-eclampsia. Participants highlighted increased monitoring and surveillance as a benefit of pre-eclampsia risk screening in addition to preventive medicines such as aspirin.

**Tool performance.** Disagreement with the original sensitivity and specificity targets was also identified as a major theme. Participants across settings suggested increasing the original proposed specificity target of >60%, as this would

reduce health system demands caused by high false positive rates, encouraging uptake. Many participants felt the original minimum sensitivity target of >60% was low, though there was mixed feedback on whether sensitivity or specificity should be higher. Participants from LMICs emphasized that for a screening test, higher specificity was critical to avoid false positives and not overburden health services. They also pointed out the high prevalence of many pre-eclampsia risk factors, so a "rule in" test, with higher specificity than sensitivity, was important.

**TPP structure.** The original draft TPP was scoped broadly covering multiple technologies, including blood biomarkers, imaging tests, biophysical measurements, and multiparametric algorithms. This broad, non-specific approach was considered challenging for product developers. For example, the minimum target for environmental stability referenced "storage and operation in a wide range of climatic conditions", which would benefit from more specific, quantifiable values. Relatedly, some participants requested a separate section for digital tools, so that digital-specific variables (e.g., design and functionality) could be specified.

**Minor themes.** Minor themes included the need for biodegradable materials, and not just recyclable materials, which was strongly suggested across participants from LMICs, given that recycling methods may not be advanced or established in some settings. Mixed feedback was given regarding the variables 'sample types, collection and processing'. For this variable's minimum target, some expressed concern that "minimal sample processing which does not require laboratory or refrigeration" was unrealistic. In contrast, others defended it, as it could help ensure equitable implementation of risk screening tools for limited-resource settings. Some participants disagreed with the variable's optimistic target (no sample required) which was seen as too ambitious.

### International stakeholder survey and public consultation

The survey was conducted between 5 June and 5 December 2024. Eighty-four survey responses were received, with representation from all WHO regions (Table 1). Participants from LMICs (LIC, LMIC and UMIC) accounted for 53.6% of responses. There were gender diversity and representation from different country income levels among respondents. Sixteen participants selected other to describe their area of expertise, citing the following: general practitioner, community health promoter, global health program developer, project implementation monitoring, project manager (x2), professional society representative (x2), international public health expert, civil society organisation representative, communications professional, geneticist, educator and sonographer. Nine respondents disclosed either a commercial or financial conflict of interest (COI) in the development of pre-eclampsia risk screening tools.

Survey results were aligned with findings from stakeholder interviews (S6 Appendix). The majority of the 25 TPP variables and their targets reached consensus: 24 of the minimum targets and 24 of optimal targets (Fig 2). The minimum targets for variables 'tool recommendations' and optimal targets for variables 'sample types, collection and processing' did not meet consensus. In addition, although meeting consensus, there was a large number of respondents who strongly disagreed with the minimum target for 'clinical specificity and sensitivity'. As a sensitivity analysis we excluded responses from those with a commercial or financial COI. This found two additional variables that did not meet consensus, namely the minimum and optimal target for 'external support' and the optimal target for 'packaging'. Sensitivity analysis excluding responses from participants in HICs found only optimal targets for variables 'sample types, collection and processing' did not meet consensus.

For the minimum target of 'clinical specificity and sensitivity', free-text comments in the survey aligned with the feedback from the individual stakeholder interviews. While both higher sensitivity and specificity targets were suggested, respondents from HICs and LMICs were concerned as to the health system burden of identifying too many women as high risk, particularly if a test produced many false positives. For 'tool recommendations', respondents stated that clinical care pathway recommendations should be a minimum requirement, due to the ethical implications of providing risk screening without subsequent guidance. For 'sample types, collection and processing', some respondents commented that the optimal target of 'no sample required' was too idealistic.

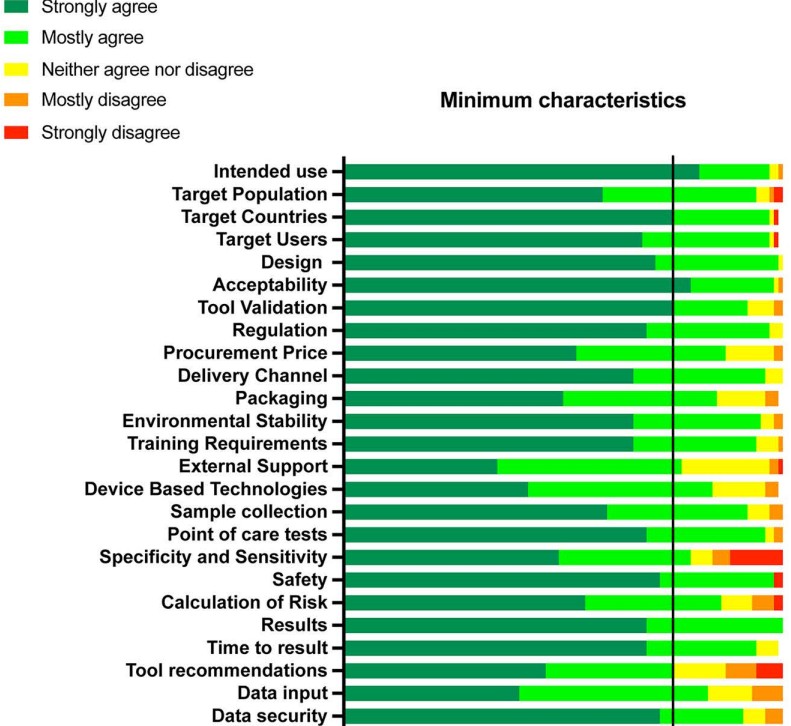

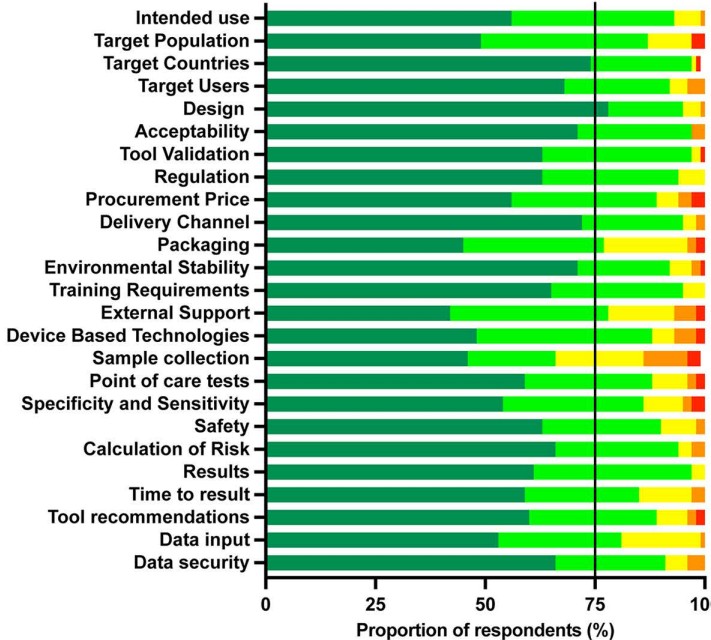

**Fig 2. Survey findings.** Consensus was considered agreement greater than 75% (black line).

Some respondents (all from HICs) were doubtful that an effective tool could exist without using invasive blood tests and ultrasounds. Whereas respondents from LMICs reiterated that urine-based biomarker testing would be ideal for their settings.

### Analysis, synthesis and finalisation

Following analysis of stakeholder feedback, the project team revised the use case scenario to ensure that the TPP addressed both preterm and term pre-eclampsia. The focus on prevention, rather than diagnosis of pre-eclampsia was also clarified.

The TPP was reorganised into three separate sections to more clearly articulate tool characteristics, tool components and data management requirements. Tool components were further separated into three categories: biochemical, bio-physical and digital components. This enabled individual test types to be clearly demarcated, though not all risk screening tools would require all test types.

Specific variables were added or amended, aligning with the feedback received across all three tool components. For example, the sensitivity and specificity targets were both increased, with specificity higher than sensitivity – this favours a 'rule-in' test. Within the biochemical tool components, the optimal target of the sample types, collection and processing variable was replaced with 'non-invasive sample required', specifying the need for minimal equipment and sample processing, and no refrigeration. Specific temperature and humidity values were added to better inform product development. Full details of the final TPP can be found in S7 Appendix.

## Discussion

### Main findings

Using a mixed methods approach and expertise from a range of international stakeholders, we have developed the first TPP for pre-eclampsia risk screening tools. Our study achieved consensus amongst participants for most TPP variables. Participants strongly agreed that 1) the scope of the TPP needed to address both preterm and term pre-eclampsia, 2) target population should be women without diagnosed pre-eclampsia and 3) increased antenatal monitoring as a benefit of screening was important. Major changes to the structure of the TPP were made in response to stakeholder feedback that the targets for some variables were too broad and lacked sufficient guidance for product developers. These changes also allowed for clearer distinction of the digital tool components. Most participants were in favour of increasing the clinical sensitivity and specificity so that the tool would function as a 'rule-in' test, to minimise overburdening the health system and increase uptake.

### Interpretation, in light of known evidence

Although less widely used than for other health conditions, some TPPs for diagnostic tools exist for pregnant women, such as for haemoglobinometers for use in antenatal care settings to diagnose anaemia [33], infectious disease diagnostics that are particularly important for pregnant women, such as Zika virus [34], and the recently published TPP for objective blood loss measurement tools to detect postpartum haemorrhage [35]. In 2018, Mugambi et al. described the end-to-end framework for introducing new diagnostic tests to limited-resource settings [36]. This framework, informed by FIND, USAID, The International Diagnostic Centre, and initiatives to introduce new TB and HIV diagnostics, describes a phase-gate model to guide product developers from conception and early phase development, through to implementation. There are five phases within the framework – 0) Concept and research, 1) Product feasibility, definition and planning, 2) Design, development and transfer to manufacturing, 3) Validation, regulatory approvals and first launch and 4) Post-launch surveillance and stable operation [36]. The TPP developed during this project aligns with this framework and includes key targets for successful progress of pre-eclampsia risk screening tools through all five stages, including starting with defining the healthcare needs, and describing critical product pre-requisites, clinical performance targets, and regulatory and safety

requirements. Diverse stakeholders with experience in target settings are critical to developing realistic and usable TPPs in LMICs. Our targeted stakeholder groups also align with the Mugambi framework, including end-users, policymakers, advocacy groups, technical experts and implementers. While procurement experts are defined as a key stakeholder in the framework, and we had initially included them in our target list of experts, currently, in many countries, pre-eclampsia risk screening is conducted based on maternal history alone, thus no procurement is required. This may contribute to the lack of stakeholders with specific expertise in procuring risk screening tools for pre-eclampsia. As point-of-care tests and other novel technologies become more available, validated and implemented globally, procurement will become more critical in pre-eclampsia risk screening and prevention.

## Research implications

The 2024 AIM project landscape analysis of the R&D pipeline of pre-eclampsia risk screening and diagnostic candidates identified 89 potential risk screening tools at various stages of development [20]. Our TPP provides a universal framework to measure the efficacy and feasibility of current and future pre-eclampsia risk screening tools, helping to promote the development of those that meet the priorities of pregnant women in limited-resource settings. Despite a large number of candidate tests undergoing R&D, the vast majority were in early-phase development. Tests that show the most promise and meet the TPP targets should be prioritised for analytical and clinical validation studies in diverse populations, as well as research on the acceptability, feasibility and cost-effectiveness, to ensure uptake.

Several candidates in the pipeline utilised machine learning algorithms [20], and emerging technologies for predicting pre-eclampsia may increasingly include artificial intelligence (AI) software. AI-based models have produced improvements in healthcare delivery and outcomes for various health conditions [37], including triaging, diagnosis and personalised risk prediction [38–40]. AI has been used in maternal and neonatal health for monitoring, early prediction and pain assessment [41]. AI-based models may have significant potential to improve the accuracy of pre-eclampsia risk screening tools, including as equitable solutions in limited-resource settings where healthcare access is limited [41]. However, accuracy of AI-based models requires large and diverse datasets, including data from LMICs to minimise bias and enhance generalisability [42]. Ethical challenges, such as data security, must also be considered to carefully assess the benefits and risks of AI in pre-eclampsia risk prediction.

## Clinical and policy implications

Innovation in health care commodities and diagnostics is often driven by laboratory discoveries or new technology, without consideration of the healthcare systems or contexts for implementation [43–45]. Only 10% and 8% of new diagnostic tests move from accuracy studies, to assessment of clinical utility and cost-effectiveness, respectively [46]. This in part leads to new diagnostics often falling into the "implementation gap". Even in the case of tests proving to be highly effective in controlled conditions, implementation can remain challenging. Consider the FMF pre-eclampsia risk calculator. While it is the recommended screening tool from international bodies, such as The International Federation of Obstetrics and Gynecology (FIGO) [47] and the International Society for the Study of Hypertension in Pregnancy (ISSHP) [48] and is used variably by clinicians in different settings, no country has yet successfully implemented it nation-wide; no national obstetric guidelines yet recommend the full FMF for all women. The "implementation gap" is even wider when considering the needs of women in LMICs. The current system often involves trying to retrofit innovative diagnostics, developed in HICs, to settings that were never considered in the development of the product. Implementation of new diagnostics into LMIC settings face unique challenges, including infrastructure and workforce constraints, procurement challenges due to reliance on imported tests or reagents, access issues due to high costs and geographic, gender and social barriers that limit care-seeking [49]. In addition, ethical implications must be considered when introducing new diagnostic tests in settings with limited or inconsistent referral pathways, preventive therapy, and health system capacity. Thus, women in LMICs continue to be left behind. TPPs are designed to address an unmet clinical need, by using the "Quality by design" framework

[50]. "Quality by design" ensures that product development uses end-to-end thinking from the start. TPPs designed to meet the needs of women in LMICs means that products will be developed to solve the real-world problems and implementation challenges in these contexts [51]. Risk screening tests designed to meet the pre-specified targets described in this TPP will have a greater likelihood of adoption and impact in clinical settings.

## Strengths and limitations

Our TPP for pre-eclampsia risk screening tools accommodates both single and multiparametric methods, and multiple test types, including blood and urine biomarkers, imaging tests and digital components. Therefore, this document can be used to guide the development of a range of risk screening tools [20]. This TPP was developed in consultation with stakeholders from ten different professional backgrounds, ensuring diversity in perspectives, including from researchers, clinical professionals, product developers and women with lived experience. This was particularly important as understanding the needs of the end users is integral for uptake and acceptance of risk screening tools. Furthermore, we ensured that there was balance across gender, country income level and WHO region amongst stakeholders. Although participation in the survey from South-East Asian representatives was limited despite the region's high disease burden, two stakeholder interviews from India (approximately 8% of interviews) provided in-depth input and informed the final TPP. The study also achieved a high response rate from other LMICs, including Africa, another region with substantial disease burden. As a result, the findings remain informative for understanding experiences and practices in high-burden settings, although caution may be warranted when extrapolating results specifically to the broader South Asian context.

Unlike TPPs for infectious disease diagnostics that target specific antigens [52,53], developing a TPP for pre-eclampsia risk screening tools presented unique complexities, owing to the evolving range of biomarkers and tests used for pre-eclampsia risk screening. The diversity of potential test types required our TPP to balance the level of detail provided for each test (e.g., specific blood test) versus the number of test types that could be included. Major structural changes were made to the TPP to incorporate sufficient detail without limiting its real-world applicability. These complexities must be considered to avoid misalignment of pre-eclampsia risk screening tools and the needs of the target population, misallocation of resources, delays and unnecessary costs, all of which will thwart progress in R&D for maternal health.

Despite the increase of AI-based models in risk prediction, this TPP did not include variables to assess this domain. This decision was widely discussed within the AIM research team, with implications around the costs and complexity of obtaining large data sets and refining the models being too complex to include within our scope. However, the exclusion of this domain should not undermine the value and benefits that AI-based models may provide for pre-eclampsia risk screening. Future TPPs for pre-eclampsia risk screening will greatly benefit from outlining specifications for AI-based risk prediction models.

## Conclusion

New risk screening tools for pre-eclampsia are urgently needed for women in LMICs, where pre-eclampsia remains a leading cause of maternal death, and easy to access preventive therapies are most often not offered to women. Through extensive stakeholder consultation, we have developed the first TPP for pre-eclampsia risk screening tools for pregnant women globally. This novel TPP will not only facilitate the development of future pre-eclampsia risk screening tools but will provide the opportunity to assess their effectiveness and impact. This could be further investigated in future by using the TPP as a framework to assess the potential risk screening candidates in the pipeline [20].

## Supporting information

**S1 Appendix. Strategies to identify stakeholders from each group.**
(DOCX)

**S2 Appendix. Interview guide.**
(DOCX)

**S3 Appendix. Survey guide.**
(DOCX)

**S4 Appendix, Countries represented by stakeholder groups.**
(DOCX)

**S5 Appendix. Coding analysis framework.**
(XLSX)

**S6 Appendix. Complete data set from survey responses (de-identified).**
(XLSX)

**S7 Appendix. Target Product Profile for Pre-eclampsia risk screening tools.**
(DOCX)

## Author contributions

**Conceptualization:** Annie McDougall, Kate Mills, Ahmet Metin Gülmezoglu, Anne Ammerdorffer, Marta Fernández-Suárez, Joshua P. Vogel.

**Data curation:** Annie McDougall, Tahlia R. Guneratne, Kate Mills.

**Formal analysis:** Annie McDougall, Tahlia R. Guneratne, Kate Mills, Maureen Makama, Ahmet Metin Gülmezoglu, Anne Ammerdorffer, Lindsay Keir, Cecile Ventola, Jennifer Scott, Marta Fernández-Suárez, Joshua P. Vogel.

**Funding acquisition:** Annie McDougall, Ahmet Metin Gülmezoglu, Anne Ammerdorffer, Joshua P. Vogel.

**Methodology:** Annie McDougall, Ahmet Metin Gülmezoglu, Anne Ammerdorffer, Marta Fernández-Suárez, Joshua P. Vogel.

**Project administration:** Anne Ammerdorffer.

**Resources:** Lindsay Keir, Cecile Ventola, Jennifer Scott.

**Supervision:** Annie McDougall, Maureen Makama, Ahmet Metin Gülmezoglu, Marta Fernández-Suárez, Joshua P. Vogel.

**Visualization:** Tahlia R. Guneratne.

**Writing – original draft:** Annie McDougall, Tahlia R. Guneratne.

**Writing – review & editing:** Annie McDougall, Tahlia R. Guneratne, Kate Mills, Maureen Makama, Ahmet Metin Gülmezoglu, Anne Ammerdorffer, Lindsay Keir, Cecile Ventola, Jennifer Scott, Marta Fernández-Suárez, Joshua P. Vogel.

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
