## [Decision Letter · Decision Letter 0]

19 Jan 2026

PGPH-D-25-03890

Expert consensus on Pre-Eclampsia Risk Screening Tools for Low- and Middle-Income Countries: Development of a new Target Product Profile

Dear Dr. McDougall,

Thank you for submitting your manuscript to PLOS Global Public Health. After careful consideration, we feel that it has merit but does not fully meet PLOS Global Public Health’s publication criteria as it currently stands. Therefore, we invite you to submit a revised version of the manuscript that addresses the points raised during the review process.

We look forward to receiving your revised manuscript.

Kind regards,

Shiyam Sunder, MBBS, MSc epidemiology, PhD

Academic Editor

Journal Requirements:

1. In the ethics statement in the Methods, you have specified that verbal consent was obtained. Please provide additional details regarding how this consent was documented and witnessed, and state whether this was approved by the IRB.

i. Please clarify all sources of financial support for your study. List the grants, grant numbers, and organizations that funded your study, including funding received from your institution. Please note that suppliers of material support, including research materials, should be recognized in the Acknowledgements section rather than in the Financial Disclosure.

ii. State the initials, alongside each funding source, of each author to receive each grant. For example: "This work was supported by the National Institutes of Health (####### to AM; ###### to CJ) and the National Science Foundation (###### to AM)."

iii. State what role the funders took in the study. If the funders had no role in your study, please state: “The funders had no role in study design, data collection and analysis, decision to publish, or preparation of the manuscript.”

iv. If any authors received a salary from any of your funders, please state which authors and which funders.

3. Please provide separate figure files in .tif or .eps format.

4. We have noticed that you have uploaded Supporting Information files, but you have not included a list of legends. Please add a full list of legends for your Supporting Information files after the references list.

5. We note that your Data Availability Statement is currently as follows: All data is included within the manuscript and supplemental materials.

Additional Editor Comments (if provided):

Reviewers' comments:

Reviewer's Responses to Questions

**Comments to the Author**

1. Does this manuscript meet PLOS Global Public Health’s publication criteria?

Reviewer #1: Yes

Reviewer #2: Yes

Reviewer #3: Yes

2. Has the statistical analysis been performed appropriately and rigorously?

Reviewer #1: Yes

Reviewer #2: Yes

Reviewer #3: I don't know

3. Have the authors made all data underlying the findings in their manuscript fully available (please refer to the Data Availability Statement at the start of the manuscript PDF file)?

Reviewer #1: Yes

Reviewer #2: Yes

Reviewer #3: Yes

4. Is the manuscript presented in an intelligible fashion and written in standard English?

Reviewer #1: Yes

Reviewer #2: Yes

Reviewer #3: Yes

Reviewer #1: 1. More details on the implementation challenges, acceptance, and sustainability (three sections wise), both at facility level and at community level.

2. South-East Asia has a high burden, however, very minimum participants took part from this region. The author should address this issue, with some possible contributing factors.

3. The author should include fetal complications (IUGR, stillbirth) associated with pre-eclampsia in introduction.

4. Please check the spelling (typo) in abstract (conclusion)

Reviewer #2: The manuscript would benefit from clarity on how LMIC stakeholder input influenced the final TPP. Which specific variables or targets were modified specifically based on feedback from LMIC participants? It would also be helpful to understand whether and how LMIC stakeholders were involved beyond interviews and surveys (e.g., in study design, interpretation, advisory roles, or review of the final TPP).

Given the heterogeneity within regions, especially within LMIC contexts it may be helpful to provide more detail on subregions or contextual settings represented to better reflect this diversity.

Finally, the manuscript may benefit from a deeper discussion of the ethical implications of introducing pre-eclampsia risk screening tools in settings where referral pathways, timely access to preventative treatment, and health system capacity may be limited or inconsistent.

Reviewer #3: i. The manuscript details the consensus methodology and agreement threshold. Could the authors clarify how disagreements were managed when variables did not reach consensus, how minority perspectives were integrated, and whether responses from different stakeholder groups were weighted equally or some prioritised during consensus determination?

ii. Can the authors clarify how the qualitative analysis was conducted in practice, including the number of coders involved, how coding disagreements were resolved, and whether thematic saturation was assessed?

iii. Given that an exact response rate could not be calculated, can the authors discuss the potential implications of non-response bias on the consensus findings and how this may affect interpretation?

iv. Given the heterogeneous stakeholder groups represented (Table 1), were all survey and interview responses weighted equally during consensus determination, or were certain stakeholder perspectives prioritised for specific variables?

v. Did the authors assess whether perspectives from HIC-based stakeholders disproportionately influenced consensus outcomes, particularly for variables related to performance targets and implementation feasibility?

vi. It may be helpful for the authors to consider including the full interview guide and complete survey instrument as supplementary materials to improve transparency and reproducibility.

vii. Can the authors carefully proofread the manuscript to address minor typographical errors and grammatical inconsistencies? e.g., “caare ” in abstract

Overall, the topic is relevant to global maternal health and aligns with the scope of PLOS Global Public Health. The study was adequately conducted and clearly reported; however, minor clarifications are required regarding the consensus process and the integration of stakeholder inputs. Overall, the manuscript is suitable for publication following minor revision.

**Do you want your identity to be public for this peer review?** For information about this choice, including consent withdrawal, please see our Privacy Policy

Reviewer #1: No

Reviewer #2: No

Reviewer #3: **Yes:** Appiah Elvis Angelo

---

## [Editor Report · Decision Letter 1]

4 Feb 2026

Expert consensus on Pre-Eclampsia Risk Screening Tools for Low- and Middle-Income Countries: Development of a new Target Product Profile

PGPH-D-25-03890R1

Dear Dr McDougall,

We are pleased to inform you that your manuscript 'Expert consensus on Pre-Eclampsia Risk Screening Tools for Low- and Middle-Income Countries: Development of a new Target Product Profile' has been provisionally accepted for publication in PLOS Global Public Health.

Best regards,

Shiyam Sunder, MBBS, MSc epidemiology, PhD

Academic Editor